# Profit Allocation Problem and Algorithm of Network Freight Platform under Carbon Trading Background

**DOI:** 10.3390/ijerph192215031

**Published:** 2022-11-15

**Authors:** Changbing Jiang, Jiaming Xu, Shufang Li, Yulian Fei, Yao Wu

**Affiliations:** 1Modern Business Research Center of Zhejiang Gongshang University, Key Research Institute of Humanities and Social Sciences for Universities, Ministry of Education of China, Hangzhou 310018, China; 2School of Management and E-Business, Zhejiang Gongshang University, Hangzhou 310018, China; 3School of Accounting and Finance, Zhejiang Vocational College of Commerce, Hangzhou 310053, China

**Keywords:** network freight, division of labor, distribution of benefits, division of labor in ant colonies, relative deprivation

## Abstract

With the gradual popularization of carbon trading, individual carbon emission behavior will come with carbon costs, which will have a significant impact on the network freight platform carrier drivers. Therefore, in order to improve the stability within the network freight platform, this paper considers the fairness of benefit distribution among network freight carriers and aims to offset the impact of carbon cost to a greater extent by reducing the relative deprivation of the network freight platform carrier group, so as to finally realize the benign operation of network freight. This paper introduces a number of indicators such as contribution value, expectation realization degree, and relative deprivation feeling, and establishes a dynamic benefit distribution optimization model oriented by relative deprivation feeling. Based on the basic characteristics of the problem, the ant colony labor division model is extended, and the corresponding algorithm is designed to solve the problem. By introducing the contribution value, contribution rate and expected return to reset the stimulus value of the environment and the response threshold of agent, the relative deprivation sense is used to quantify the degree of unfair benefit distribution, which is used as a benchmark to dynamically coordinate the benefit distribution of the carrier group and optimize the benefit distribution scheme. The experimental results show that the extended model and algorithm designed in this paper can significantly reduce the relative deprivation perception of the carrier group in the online freight platform at a low cost.

## 1. Introduction

Carbon trading is a trading method of buying and selling greenhouse gas emission allowances, which is mainly for carbon dioxide [1]. Carbon trading is based on the carbon market, and the current carbon market in China is mainly a pilot carbon exchange, some of which can support individuals to open accounts for carbon trading [2]. At present, the more mature carbon trading mechanism is mainly for emission-controlled enterprises and determines whether the enterprises need to buy carbon credits by the difference of carbon credits. Once the credits are exceeded they need to be made up, and on the contrary, enterprises with surplus credits can sell the credits, during which all transactions are carried out in the carbon market. In this regard, in order to further promote the implementation of environmental protection policies, carbon control requirements will become more and more stringent, and corresponding carbon credits will be set for individuals. While the general population may only consume carbon credits for commuting and electricity consumption, for private truck drivers (including C2C drivers of online freight platforms), carbon credits are particularly important. The externalities caused by goods transportation are not only related to carbon emissions: Pourrahmani and Jaller [3], in their study for crowdshipping in last mile deliveries, found that there is an impact of evaluating services on service participants, which pointed out that appropriate compensation schemes can lead to high capital efficiency and sustainable operations. Silvestri et al. [4] in their study for the optimization of networked goods migration in urban stores pointed out that a shared information-based model of networked goods migration can reduce the pollution generated during transportation and thus its associated externalities. Masteguim and Cunha [5], on the other hand, conducted a study on the impact of pickup points on last-mile delivery and showed that reasonable delivery points can circumvent more waste of resources. In addition, Hailemariam et al. [6] found that income inequality is positively correlated with carbon emissions, thus achieving a fair distribution of benefits is also essential to promote a low-carbon economy.

According to statistics, in the period of 2020–2021, the total volume of freight transport in China was well over 46 billion tons and road freight accounted for over 70% of it, which shows that it is still the main mode of transport for freight transport in China. Along with the widespread use of the Internet and digital technology in China, the road freight in the era of big data-network freight industry has been unprecedented development. By 2021, China’s network freight companies exceeded 1900, and its market size is as high as 338.9 billion yuan, but the problems also came one after another. Due to the rapid expansion of the market scale in a short period of time, the benefit distribution mechanism within the platform is not yet perfect, and combined with the various carbon control policies that will be implemented soon, this will affect the stability within the platform to a certain extent. Because there are interests in the distribution of orders in the network freight platform, and there may be deviations between the expected and actual interests of each carrier, as well as the need to offset the impact of carbon costs [7], it is very easy to produce a sense of relative deprivation [8] leading to a reduction in the internal stability of the enterprise. The sense of relative deprivation is a measure of the internal stability of the platform from the perspective of fairness in the distribution of benefits, so in order to minimize the impact of the above-mentioned problems, it is also necessary to coordinate the interests of the profitable parties, so this paper further studies the distribution of benefits among the groups of carriers of the network freight platform.

With the development of the Internet of Things, smart cities have become the mainstream of urbanization [9], which accomplishes high-precision information processing through the powerful scale data processing algorithm of the Internet and breaks the information barriers to integrate resources on this basis [10]. The network freight platform is a multi-party information integration platform, which matches the vehicle and cargo resources of shippers and carriers to finally meet the needs of shippers and give compensation to carriers. For the benefit distribution, different benefit distribution schemes will obviously affect the relative deprivation of the carrier group on the platform. In order to improve the internal stability of the platform as much as possible, this paper conducts a study at the level of improving the fairness of benefit distribution. As shown in Figure 1, this paper intends to centrally regulate the interests of carrier drivers through the online freight platform, and to drive the benefit distribution to be as fair as possible through the evaluation of relevant indicators. Specifically, the problem studied in this paper is the calculation of the relative deprivation of the group of carriers with the known initial benefit distribution scheme, and the dynamic coordination of benefit redistribution by using this as a measurement indicator, in order to maximize the fairness of benefit distribution without affecting the economic efficiency of the platform.

In summary, based on the combination of the ant colony labor division model and the fixed threshold response model, this paper designs an algorithm to study the benefit distribution among the carrier drivers of the online freight platform, and verifies the validity and applicability of the model and the algorithm with simulation experimental data. The paper aims to reduce the overall relative deprivation of network freight platform carriers by adjusting the benefit distribution, with the aim of improving the internal stability of the platform, thus reducing the negative impact of income inequality on carbon emissions, and providing theoretical support for subsequent related research in this field.

## 2. Literature Review

Since interest distribution involves group interests, Baker [11] defines a group as an aggregate composed of individuals with interdependent relations, whose members actually depend on each other to achieve individual and collective goals. Problems involving groups can often be solved by using swarm intelligent division of labor, which is the overall intelligent behavior of a group composed of many simple *agents* spontaneously emerging to cooperate to complete a common task. It quantifies some allocation [12] problems in a specific environment into mathematical models and uses swarm intelligence algorithm to obtain the optimal solution or approximate optimal solution.

In order to better apply the methodology to allocation in these different environments, many swarm intelligence algorithms are derived to achieve more reasonable allocation. Bonabeau [13] et al. proposed a fixed response threshold model (FRTM) when improving ant colony algorithm, which greatly enhanced the positive feedback mechanism of ant colony algorithm, but also had the problem that it was difficult to find the appropriate value of threshold, and all ants’ threshold values were the same and remained unchanged, which had certain limitations. Therefore, Bonabeau [14] and others based on ant colony distributed control were put forward based on the stigmergic communication paradigm of ant colony algorithm, using different forms of incentive variables by encouraging communication.

At present, there are two kinds of profit distribution according to income value distribution and cost distribution. Dai Jianhua and Xue Hengxin [15] used the Shapley value model to study the benefit distribution of partners in dynamic alliance (virtual enterprise), and the results showed that this method could avoid simple average distribution and improve the enthusiasm of alliance members. Dai Bo [16] pointed out that in collaborative logistics, carriers can improve resource utilization by sharing vehicle and cargo resource information. He proposed three profit distribution mechanisms based on Shapley value, proportional distribution concept and contribution for the pickup and delivery service problem (CCPPD). Zhao Wenjian and Liu Jiacai [17] proposed a solution model of a cooperative game based on least square method considering the existence of cooperative games in the transportation process of road freight alliance, and verified through numerical experiments of benefit distribution that the benefit of road freight cooperation is greater than that of working alone. In addition, Liu Jiacai et al. [18] also studied the distribution of benefits in logistics coordination alliance with incomplete information and established an extended model based on the profits and contributions of member enterprises. The results showed that this distribution method could improve the overall distribution fairness and efficiency. Kumoi and Matsubayashi [19] analyzed vertical integration in supply chain under cooperative game from the perspective of profit distribution, and the results showed that vertical integration was stable when members were pessimistic. Wang Yong et al. [20] proposed a dual-objective mixed integer programming model based on the state-spatio-temporal network to minimize the operating cost and the number of distributed vehicles by combining the vehicle routing problem and the profit distribution problem. The results show that this method can achieve overall cost reduction and efficiency increase of freight transportation.

Based on this, this paper takes the revenue of the network freight platform carriers in the supply chain as the optimization objective, and takes the contribution, expected benefit and relative deprivation as the measurement index in order to dynamically coordinate their benefit distribution problem.

## 3. Overview of Problem Cases and Methods and Theories

### 3.1. Problem Background

In the process of transportation order distribution from the beginning of the supply chain to the complete end of order delivery, among the network freight platform carrier drivers, there are those who pay a relatively high return, and there are also those individuals who cause their own income to be not ideal in order to protect the reputation of the network freight platform. In this case, the platform should reasonably redistribute the interests among the driver groups, and subsidize the latter if necessary, so as to make them satisfied and continue to obey the order dispatch of the platform. On the whole, after the order distribution is completed and the order transportation is completely finished, the expected benefit realization degree of each carrier driver is inconsistent in the whole process, which will lead to the improvement of the relative deprivation sense. When the relative deprivation sense of the group is significantly increased, the internal stability of the platform will be reduced. In other words, after completing the order distribution, it is necessary to regulate and distribute the interests of the carrier group of the network freight platform, and then maintain healthy operation inside the platform.

### 3.2. Profit Distribution and Related Indicators

There are many forms of interest. Desai [21] proposed that interest can be an additional form product such as value, rights and relations, but this explanation still cannot fully summarize it. Battistini [22] pointed out that benefits do not come from the scarcity of a certain form of products, but from the collective nature of the value creation process, which is easily produces the conflict of low efficiency in the stage of income generation and distribution. As for the distribution of benefits, it can be regarded as either dynamic or static equilibrium, depending on the specific scenario of the distribution of benefits. However, in either scenario, the income of interest distribution is more concerned about the realization degree of expected benefits and the transformation degree of contributed benefits.

In the problem of interest distribution, there are multiple interest subjects, and the interests created by them constitute the overall interests. However, due to the interest distribution in the group, individuals in the group will make different contributions due to their own characteristics, which also leads to the differences in expected income among individuals. The expression of contribution is as follows:(1)$$C_{i}=\underset{j}{\sum }d_{j}\cdot q_{ij}$$
where, *C_i_* represents the contribution value made by *agenti* in the process of order allocation, *q_ij_* represents the input of *agenti* in *j* index and *d_j_* is the corresponding weight coefficient of each index. It can be seen that the contribution value is directly proportional to the input.

The contribution rate is expressed by the proportion of the input of interest subjects in the whole group, as follows:(2)$$c_{i}=C_{i}/\sum C_{i}$$

In the formula, $\sum C_{i}$ represents the total contribution value made by all interest subjects in the process of order distribution. It can be seen that the contribution rate represents the contribution ratio of interest subjects to some extent. Since the total contribution value is constant after the completion of the whole division of labor, *c_i_* is mainly determined by the size of *C_i_*.

The profit in the model refers to the profit obtained by the interest subject in the whole process of division of labor. Generally, this value is given after the labor distribution is completed and *G_i_* is used for expression. The expected profit is equal to the product of the contribution rate and the total benefit of the group, and the specific expression is as follows:(3)$$Q_{i}=c_{i}\cdot \sum G_{i}$$
where *Q_i_* is used to represent the expected return of *agenti*, $\sum G_{i}$ represents the total return produced by the group in the whole process of division of labor. Since the total return obtained by the group after labor distribution is a fixed value, it can be seen that the expected return *Q_i_* of *agenti* is mainly determined by the contribution rate.

Through the calculation of (1), (2) and (3), the *A_i_* is realization degree of *agenti* expected income can be further calculated, and the relative deprivation *RD_i_* of each *agent* can be calculated according to the calculation *A_i_* as follows:(4)$$A_{i}=(\frac{G_{i}}{Q_{i}})\cdot 100\%$$
(5)$$RD_{i}=AD\left(A_{i}\right)\cdot P\left(A_{i}\right)$$
where the realization degree of expected revenue *A_i_* is equal to the value *G_i_* divided *Q_i_* and multiplied by 100%, while on the right side of the relative deprivation *RD_i_* equation, $AD\left(A_{i}\right)$, $P\left(A_{i}\right)$ respectively represent the average of the realization degree of expected revenue of *agent* with greater value and the proportion of *agenti* with higher value in the whole group. It can be seen that the generation of relative deprivation is caused by the unfairness of benefit distribution. When the expected benefit realization degree *A_i_* of all *agent* can be equal, *RD_i_* will reach a minimum value, and the benefit distribution at this time is the optimal allocation scheme for the ant colony division of labor model of group benefit distribution.

### 3.3. Swarm Intelligence Benefit Allocation Method

This problem of interest distribution is essentially a problem of division of labor, which is defined by Durkheim [23] as an activity of complex division of labor for common and mutually indispensable tasks of a group with different properties. Due to the different contributions of each participant when completing the task together, their expected benefits will also be different. After the actual distribution of benefits is completed, an expected degree of benefit realization can be calculated. Once the difference of the degree of expected benefit realization between groups is too large, the enthusiasm of the whole group will be reduced. Therefore, it is necessary to make a reasonable benefit distribution scheme for the benign development of a benefit group.

Based on the combination of Bonabeau and other crowd intelligence studies, Xiao Renbin et al. [24] coordinated the benefit distribution among groups based on the perspective of cost payment, by qualitatively analyzing the benefit distribution and quantifying the contribution, benefit, contribution ratio and expected benefit in the benefit distribution in the model. The relative deprivation sense is introduced to measure the fairness of benefit distribution, and finally, the influencing factors in these problems are combined with the ant colony labor division model to construct a group benefit distribution model based on ant colony labor division. In addition, He Zhengzang and Huang Juan et al. [25] introduced a penalty factor to punish the loss caused by agent’s interruption or withdrawal from performing tasks on the original basis, and also introduced relative deprivation sense to improve the dynamics and fairness of multimodal transportation benefit distribution, and the results indicated that the model could effectively improve the equilibrium of enterprise earnings.

### 3.4. Profit Distribution Model Based on Ant Colony Division of Labor

In the interest distribution, since the total revenue is constant, once the profit of the interest subject increases, the profit of the interest subject will inevitably decrease. For the environmental stimulus in the interest distribution model, there are two types, namely, the environmental stimulus value under the two conditions of interest increase and interest decrease respectively, and the specific expression is as follows:(6)$$S_{i}{}^{+}=C_{i}/\left(C_{i}+G_{i}\right)$$
(7)$$S_{i}{}^{-}=G_{i}/\left(C_{i}+G_{i}\right)$$

When increasing the interest allocated by *agenti*, consider that it is in a state where the contribution is more than the return is less. When *C_i_* > *G_i_*, it is chosen as the sum of the numerators divided by *C_i_*. On the contrary, when the benefit allocated by *agenti* is reduced, it is considered to be in a state where the contribution is less and the return is more. In this case, it is chosen as the sum of the numerator divided by *G_i_*.

According to Equation (6), when *G_i_* invariant, $S_{i}{}^{+}$ is proportional to the value of *C_i_*; when *C_i_* invariant, $S_{i}{}^{+}$ is inversely proportional to the value of $G_{i}$; similarly, for (7), when $G_{i}$ invariant, $S_{i}{}^{-}$ is inversely proportional to the value of *C_i_*, and when *C_i_* invariant, $S_{i}{}^{-}$ is inversely proportional to the value of *G_i_*.

In the interest distribution, response threshold represents the interest realization ability of the interest subject, which can be embodied in resources, capabilities and means, etc. In view of the above environmental stimulus values of interest increase and interest decrease, response threshold values of interest increase and interest decrease are respectively set, and the specific expressions are as follows:(8)$$\theta_{i}{}^{+}=1/\left(k\cdot E_{i}\right)$$
(9)$$\theta_{i}{}^{-}=k\cdot E_{i}$$

In the process of profit distribution, the *agent* with stronger ability of profit realization will desire more benefits. On the contrary, the *agent* with weaker ability to realize benefits will also desire less benefits. Therefore, when the *E_i_* is more, the larger the possibility of increasing the response benefit is. On the contrary, when the *E_i_* is less, the smaller the possibility of increasing the response benefit is. In order to simplify the model, Xiao Renbin et al. assume that $\theta_{i}{}^{+}$ is inverse relationship to *E_i_*, and $\theta_{i}{}^{-}$ is direct relationship to *E_i_*. In Equations (8) and (9), *k* is the weight coefficient of the range of regulating threshold.

For stakeholders in the process of profit distribution, there are three kinds of behavior choice—interests increase, decrease and interests are the same—so the *agent* distribution of interests in the process with time there are three possible response probabilities, namely benefit increases the probability that the response is $P_{i}{}^{+}$, benefit reduces the probability of the response being $P_{i}{}^{-}$, and the response to the interests of the same probability is $P_{i}{}^{*}$. whose expressions are as follows:(10)$$P_{i}{}^{+}=\frac{{\left(S_{i}{}^{+}\right)}^{2}}{{\left(S_{i}{}^{+}\right)}^{2}+{\left(\theta_{i}{}^{+}\right)}^{2}}$$
(11)$$P_{i}{}^{-}=\frac{{\left(S_{i}{}^{-}\right)}^{2}}{{\left(S_{i}{}^{-}\right)}^{2}+{\left(\theta_{i}{}^{-}\right)}^{2}}$$
(12)$$P_{i}{}^{*}=1-P_{i}{}^{+}-P_{i}{}^{-}$$

The response probability formula in the ant colony labor division fixed threshold response model is followed here, with the slight difference that different variables are set up to calculate the corresponding response probability for different behavioral choices. It can be seen that the probability of increasing *agent* interest is directly proportional to $S_{i}{}^{+}$ and inversely proportional to $\theta_{i}{}^{+}$; the probability of decreasing *agent* interest is also directly proportional to $S_{i}{}^{-}$ and inversely proportional to $\theta_{i}{}^{-}$; the probability of unchanged *agent* interest is jointly determined with both $P_{i}{}^{+}$ and $P_{i}{}^{-}$.

After the agent chooses the behavior of interest change, it needs to adjust the interest. The specific formula is as follows:(13)$$\varepsilon_{i}=H_{1}\cdot S_{i}{}^{+}\cdot E_{i}\cdot \mathrm{rand}\left(\xi_{1}\right)-H_{2}\cdot S_{i}{}^{-}\cdot \left(1/E_{i}\right)\cdot \mathrm{rand}\left(\xi_{2}\right)$$

In Equation (13), $H_{1}$ and $H_{2}$ are decision state variables, equal to 1 or 0. When *agent* selects benefit increases $H_{1}=1$, on the contrary $H_{1}=0$; when *agent* decreases $H_{2}=1$, on the contrary $H_{2}=0$; when *agent* does not change, $H_{1}=H_{2}=0$. Considering the positive and inverse relationship between the interest realization ability of the interest subject and the expected interest, the interest when the interest increases taking $E_{i}$ and $1/E_{i}$ when the interest decreases. $\xi_{1}$ and $\xi_{2}$ respectively represent the mean of the increase and decrease in the process of interest redistribution, $\mathrm{rand}\left(\xi_{1}\right)$ and $\mathrm{rand}\left(\xi_{2}\right)$ represent any value from the (0, $\xi_{1}$) and (0, $\xi_{2}$) intervals, respectively.

Considering that the interests of the interest subject will change in the process of benefit redistribution, the demand of the interest subject desiring benefit redistribution will gradually decrease at this time, and the expected benefits of *agent* cannot be changed due to the constant contribution paid, so the threshold of *agent* response to benefit change is regulated at this time by regulating the benefit realization ability of the interest subject, thus the specific expression is as follows.
(14)$$G_{i}^{'}=G_{i}+\varepsilon_{i}$$
(15)$$E_{i}^{'}=\{\begin{array}{c} z_{1}\cdot E_{i},\;\;G_{i}^{'}>G_{i}\\ z_{2}\cdot E_{i},\;\;G_{i}^{'}<G_{i} \end{array}$$

In Equation (14), the new income after *agenti* redistribution is equal to the benefit obtained at the initial moment $G_{i}$ plus the change of interest $\varepsilon_{i}$. In Formula (15), $E_{i}^{'}$ for the updated ability to realize the interests of the interest subject, $0<z_{1}<1,$ $z_{2}>1$. When the interest distribution of the interest subject increases, $G_{i}^{'}>G_{i}$. The ability to realize the interests of the stakeholders will be reduced. On the contrary, when the distribution of benefits decreases, the ability to realize benefits will increase.

This paper aims to promote the development of low-carbon freight transport, takes low-carbon emissions as an environmental constraint, and improves the fixed threshold response model based on the division of labor of the ant colony, thus further expanding the order allocation response model of the network freight platform in order to reduce the carbon emissions of the network freight industry.

## 4. Modeling of Order Allocation of Network Freight Platform under the Carbon Tax Policy Constraint

### 4.1. Variable Description

Considering the problem context and environment, the variable descriptions of the extended model for the distribution of benefits to drivers of online freight platforms are given in Table 1.

### 4.2. Benefit Distribution Model of the Carrier of the Network Freight Platform

There are *n agents* involved in the benefit distribution, so the research object in this paper is to obtain the benefit distribution *agent*, and the optimization target *Z* = $\left(AG_{i},\;RD_{i},\;t\;\right)$, *i* ∊ *n*, *t* represents the time variable.

(1) Contribution and contribution rate

In the preceding order distribution problem there are multiple subjects of interest, the network freight platform carrier drivers will receive the corresponding compensation after completing the order, which constitute the overall benefits of the network freight platform, but because each network freight platform carrier drivers make different contributions in the process of order distribution, so their expected benefits differ, where the contribution reference to the distribution costs in the preceding section, specifically The expression is as follows:(16)$$AC_{i}=\alpha \cdot C_{i}+\beta \cdot E_{i}\cdot w$$

In the formula, $AC_{i}$ represents the contribution value made by *agenti* in the order distribution process, $C_{i}$ represents the final transportation cost incurred by *agenti* in the order distribution problem, $E_{i}*w$ is the cost of carbon emissions and the weight coefficient corresponds to each index α and β. It can be seen that the contribution value is directly proportional to the input cost.

The contribution rate is expressed by the proportion of the input of the stakeholders in the whole group, which is as follows:(17)$$ac_{i}=AC_{i}/\sum AC_{i}$$

In this action, $\sum AC_{i}$ represents the total contribution of all the online freight platform carrier drivers involved in order delivery. It can be seen that the contribution rate represents the *agenti* contribution ratio to some extent. Since the total contribution value of the entire order delivery remains constant, $ac_{i}$ is mainly determined by the size of $AC_{i}$.

(2) Earnings and expected earnings

The benefits in the model are the benefits obtained by the network freight platform carrier drivers in the whole order distribution process. This value is given by the above order distribution model, and the expected benefits are equal to the product of the contribution rate and the total benefits of the group. The specific expression is as follows:(18)$$Q_{i}=ac_{i}\cdot TAG_{i}$$

In the formula, $Q_{i}$ is used to represent the expected benefit of *agenti* and $TAG_{i}$ represents the total revenue obtained by the online freight platform carrying drivers participating in the order delivery after the total order allocation in the previous article. Because the total return is a fixed value, it can be seen that *agenti* expected return $Q_{i}$ is mainly determined by the contribution rate and proportional.

(3) Degree of expected benefit realization and relative deprivation

After (4)–(16), (4)–(17) and (4)–(18) calculations, the expected benefit realization degree of *agenti* is further calculated. The relative deprivation sense $RD_{i}$ was also calculated for each *agent* on the basis of *A_i_*, details are as follows:(19)$$A_{i}\left(t\right)=(\frac{AG_{i}\left(t\right)}{Q_{i}})\cdot 100\%$$
(20)$$RD_{i}\left(t\right)=AD\left(A_{i}\left(t\right)\right)\cdot P\left(A_{i}\left(t\right)\right)$$

In the formula, the degree $A_{i}$ of expected return realization is equal to $AG_{i}$ divided by $Q_{i}$ and multiplied by 100%, while $AD\left(A_{i}\right)$ and $P\left(A_{i}\right)$ on the right side of the relative deprivation equation represent the mean value of the *agent* and the proportion of the higher *agent* in the whole group. It can be seen that the sense of relative deprivation is caused by the injustice of benefit distribution. When all the expected benefit realization degree of *agent* is equal, $RD_{i}$ is equal to 0. At this time, the benefit distribution is the optimal distribution scheme for the ant colony division of labor model of group benefit distribution.

In benefit distribution, because it is redistributed, the total income of the group is constant, so once the *agent* profit increases, there will be *agent* profit reduction, for the benefit distribution model of environmental stimulus is divided into two kinds, namely respectively consider the interest increase and reduce the benefits of environmental stimulus value, The specific expression is as follows:(21)$$S_{i}{}^{+}\left(t\right)=AC_{i}/\left(AC_{i}+AG_{i}\left(t\right)\right)$$
(22)$$S_{i}{}^{-}\left(t\right)=AG_{i}\left(t\right)/\left(AC_{i}+AG_{i}\left(t\right)\right)$$

At the increase of the benefits allocated to the *agenti*, it is in a state of more contribution and less return. At this time, $AC_{i}$ > $AG_{i}$, so $\;AC_{i}$ is selected as the molecule divided by the sum of $AC_{i}+AG_{i}$. In contrast, when the benefits allocated by the *agenti* are reduced, it is a state of less contribution and more return, and $AC_{i}$ < $AG_{i}$. Therefore, $AG_{i}$ was selected as the molecules divided by the sum of $AC_{i}+AG_{i}$.

(4) Benefit realization capability and response threshold

In the order distribution, the benefit realization ability is reflected in the *agent*, that is, “the greater the ability, the greater the responsibility”. Here, using the *agent*, the response threshold represents the perception degree of the benefit subject to response to the benefit change, it sets the response threshold for the response increase and benefit reduction, as follows:(23)$$AE_{i}=\rho v_{i}/\left(c_{i}\cdot \mu_{i}\right)$$
(24)$$\theta_{i}{}^{+}=1/\left(k\cdot AE_{i}\right)$$
(25)$$\theta_{i}{}^{-}=k\cdot AE_{i}$$

In the process of order transportation, the larger the vehicle *agent*, the more cargo it wants to transport, the more benefits it wants, and the more the unit transportation cost and carbon emission coefficient, the higher the consumption cost is. In conclusion, $AE_{i}$ is the proportional relationship with $v_{i}$, and the inverse relationships with $c_{i}$ and $\mu_{i}$, ρ is the benefit of the realization ability weight coefficient.

In the process of profit distribution, the more powerful the *agent* is, the more benefits it desires to get; on the contrary, the *agent* with weaker ability to realize benefits will have less desired benefits. Therefore, the larger the $E_{i}$, the greater the possibility of increasing the response benefit and the smaller the response threshold $\theta_{i}{}^{+}$ is. On the contrary, if the $E_{i}$ is greater, the possibility of increasing the response benefit is smaller, and the response threshold $\theta_{i}{}^{-}$ is more. In order to simplify the model, Xiao Renbin et al. assumed that $\theta_{i}{}^{+}$ and $E_{i}$ are inversely proportional to each other and that $\theta_{i}{}^{-}$ and $E_{i}\;$ are directly proportional to each other. In Equations (24) and (25), *k* is the transformation coefficient of the range of adjustment threshold.

(5) Response probability and benefit change

In the process of interest after allocation, the interests of the *agent* may have three changes—interests increase, decrease and interests are the same—so in the *agent* distribution of interests in the process with time, there are three possible response probabilities, namely benefit increases the probability of the response $P_{i}{}^{+}$, benefit reduces the probability of the response $P_{i}{}^{-}$ and the response to the interests of the same probability is $P_{i}{}^{*}$. The specific expression is as follows:(26)$$P_{i}{}^{+}\left(t\right)=\frac{{\left(S_{i}{}^{+}\left(t\right)\right)}^{2}}{{\left(S_{i}{}^{+}\left(t\right)\right)}^{2}+{\left(\theta_{i}{}^{+}\left(t\right)\right)}^{2}}$$
(27)$$P_{i}{}^{-}\left(t\right)=\frac{{\left(S_{i}{}^{-}\left(t\right)\right)}^{2}}{{\left(S_{i}{}^{-}\left(t\right)\right)}^{2}+{\left(\theta_{i}{}^{-}\left(t\right)\right)}^{2}}$$
(28)$$P_{i}{}^{*}\left(t\right)=1-P_{i}{}^{+}\left(t\right)-P_{i}{}^{-}\left(t\right)$$

Based on the response probability formula in the fixed threshold response model, the slight difference is that it chooses different behaviors and different variables were set up to calculate the corresponding response probabilities. It can be seen that the probability of increasing agent’s interest is directly proportional to and inversely proportional to, respectively; the probability of decreasing *agent* interest is also directly proportional to $S_{i}{}^{+}$ and $\theta_{i}{}^{+}$, and inversely proportional to $S_{i}{}^{-}$ and $\theta_{i}{}^{-}$, and the probability of unchanged *agent* interest is jointly determined with both $P_{i}{}^{+}$ and $P_{i}{}^{-}$, respectively.

After the *agent* choice of benefit change behavior, and benefits need to be regulated, the specific formula is as follows:(29)$$\varepsilon_{i}\left(t\right)=G_{i}{}^{+}-G_{i}{}^{-}$$
(30)$$G_{i}{}^{+}\left(t\right)=H_{1}\cdot S_{i}{}^{+}\left(t\right)\cdot AE_{i}\cdot \mathrm{rand}\left(\xi_{1}\right)$$
(31)$$G_{i}{}^{-}\left(t\right)=H_{2}\cdot S_{i}{}^{-}\left(t\right)\cdot \left(1/AE_{i}\right)\cdot \mathrm{rand}\left(\xi_{2}\right)$$
(32)$$G_{i}\left(t+1\right)=G_{i}\left(t\right)+\varepsilon_{i}\left(t\right)$$

In Equations (4)–(29), $\varepsilon_{i}$ represents the change value of *agenti* interest, $G_{i}{}^{+}$ is the value increase in the process of interest regulation, and on the contrary, $G_{i}{}^{-}$ is the value of decrease. In Equations (4)–(30) and (4)–(31) are decision state variables, equal to 1 or 0. When the *agent* choice benefits increase, ($P_{i}{}^{+}>P_{i}{}^{-},\;P_{i}{}^{+}>P_{i}{}^{*}$), $H_{1}=1$, and vice versa. Likewise, when the *agent* choice benefits decrease ($P_{i}{}^{-}>P_{i}{}^{+},\;P_{i}{}^{-}>P_{i}{}^{*}$), $H_{2}=1$, and vice versa, $H_{2}=0.$ When the *agent* does not change ($P_{i}{}^{*}>P_{i}{}^{+},\;P_{i}{}^{*}>P_{i}{}^{-}$), $H_{1}=H_{2}=0$. Considering the positive and negative ratio between the interest realization ability and the expected interest of the interest subject, the value of $E_{i}\left(t\right)$ is taken when the interest increases and $1/E_{i}\left(t\right)$ when the interest decreases. Since the total interest is unchanged, the adjustment value is compared with 1% of the total interest. $\xi_{1}$ and $\xi_{2}$ respectively represent the change value of the increase and decrease of interest in the process of the complex distribution of interest. It means $\mathrm{rand}\left(\xi_{1}\right)$ is that any value is taken from the interval (0, $\xi_{1}$), and $\mathrm{rand}\left(\xi_{2}\right)$ represents any value that is taken from the interval $\mathrm{rand}\left(\xi_{1}\right)$.

In Equation (32), the new income of *agenti* after redistribution is equal to the interest obtained at the initial time $\;G_{i}$ plus the change in interest $\varepsilon_{i}$.

(6) Mean and variance of group relative deprivation and variance of expectation attainment

Given that the benefit realization ability of the network freight platform carrier driver in this model is mainly determined by the vehicle model, the recovery factor is not considered in this model. Considering the interests of the interests in the process of distribution in dynamic coordination, *agent* relative deprivation has been changing in order to consider the final interests distribution coordination effect. The model proposed group of relative deprivation mean and variance as the effectiveness of the model index, specific expression is as follows:(33)$$\bar{\mathrm{RD}}\left(t\right)=\frac{\sum RD_{i}\left(t\right)}{n}$$
(34)$$\hat{\mathrm{RD}}\left(t\right)=\frac{\sum {\left(RD_{i}\left(t\right)-\bar{RD_{i}}\left(t\right)\right)}^{2}}{n}$$
(35)$$\hat{A_{i}}\left(t\right)=\frac{\sum \left(A_{i}\left(t\right)-1\right){)}^{2}}{n}$$
where $\bar{RD_{i}}$ is the average of the group relative deprivation, $\bar{RD_{i}}$ is the squared difference of the group relative deprivation, when $\bar{RD_{i}}$ > σ indicates that the relative deprivation at this time is still in the higher range, so the benefit redistribution is not yet over; on the contrary, it indicates that the relative deprivation at this time has reached the acceptable range of the network freight platform carrier group, and the benefit redistribution is completely over. At this time, the benefit distribution plan is the final plan, and the acceptability of the distribution result can be tested by $\hat{\mathrm{RD}}$, and a large $\hat{\mathrm{RD}}$ indicates that the effect is less desirable, and vice versa. $\hat{A_{i}}$ is the squared difference of the standard expected benefit realization of the group, and since the standard expected benefit realization is 1, it is used to calculate the stability of the realization, and the smaller the $\hat{A_{i}}$ becomes, the better the regulation effect is.

### 4.3. Algorithm Implementation

The flow chart of the algorithm is shown in Figure 2:

The specific steps are as follows:

Step 1, set the following variables: $AG_{i}$ called *agenti* in the above order allocation problem of the initial costs, transportation costs $C_{i}$ and carbon emissions earnings $E_{i}$, maximum bearing capacity $v_{i}$, unit costs $c_{i}$, a carbon tax *w*, coefficient of carbon emissions $\mu_{i}$, fuel cost carbon conversion coefficient *α* and conversion coefficient of *β*, take the initial simulation time *t* = 0, *t_max_* for maximum operation frequency, the conversion coefficient of setting the threshold value *k*, σ is the critical value of satisfaction with group relative deprivation.

Step 2, calculate the contribution and contribution rate of *agenti* according to Equations (16) and (17), and the expected revenue of *agenti* according to Equation (18).

Step 3, calculate the realization degree of expected income and relative deprivation according to Equations (19) and (20);

Step 4, calculate the mean and variance of the relative deprivation of the group and the variance of the degree of realization of desired benefits according to Equations (33)–(35), respectively.

Step 5, calculate the environmental stimulus value and response threshold according to Equations (21)~(22) and (23)~(25).

Step 6, calculate the choice probability of *agent* interest change according to Equations (26)–(28), and the behavior choice of *agent* interest change is made according to the principle of probability Max, and the income of *agent* is updated according to Equations (29)–(32).

Step 7, if the mean of group relative deprivation ($\bar{\mathrm{RD}}$) is > the critical value and *t* + 1 < *t*_max_, let *t* = *t* + 1, go to step 3. Otherwise, go to step 8.

Step 8, statistics and output simulation results.

## 5. Numerical Experiments and Discussion

### 5.1. Problem Description and Parameter Setting

Track parts of a large factory *X* have multiple distribution outlets in *Z* city, and each distribution outlet of the factory has corresponding supply channels. Distribution network has *n* in the city, near the factory existing *n* freight network about car (*agent*) to complete the factory order distribution, total order to participate in distribution of the shipper number *n =* 18. The results of the order distribution and vehicle scheduling are shown in Table A1 in the Appendix A. Considering that some network freight platform carrier drivers are not satisfied with the current situation of getting paid, their benefits are now redistributed using the benefit distribution extension model algorithm in order to improve the fairness of benefit distribution after order distribution as much as possible. The model parameters are as follows: the maximum number of simulations runs *Tmax =* 20, the conversion coefficient of fuel cost *α =* 1, the conversion coefficient of carbon emission cost *β =* 20, the conversion coefficient of threshold *K =* 2, the conversion coefficient of benefit realization ability ρ=2, the increment of benefit regulation $\xi_{1}=10$, the negative increment of benefit regulation $\xi_{2}=5$ and the critical value of satisfaction of group relative deprivation σ = 0.1. The relevant data of each vehicle type, initial benefit income and benefit realization ability are shown in Table 2:

### 5.2. Interpretation of Result

According to the above model, the simulation experiment in this paper is carried out in the MacBook Air notebook with Intel(R) Core(TM) i5 CPU processor and 4 GB (RAM) memory. The experimental environment is the Window7 flagship operating system, and MATLAB R2018A programming is used to implement the algorithm. The running results are as follows:

Table 3 shows the initial data of each agent when the benefit redistribution is not carried out and the final data when the benefit redistribution is finally completed. The relative deprivation of *agent*5 to *agent*11 is more than 0.6. At the same time, the large difference in the degree of expected benefit realization also indirectly leads to a higher overall relative deprivation of the agent group as well. Combined with Figure 3 and Figure 4, it can be seen that after the benefit reallocation process regulates the *agent*’s benefit, although the *agent*’s benefit fluctuates less, the final value of the final degree of realization of the *agent*’s expected benefit is within the interval of 1 ± 0.05, which indicates that the model can stably control the final benefit of reallocation between 95% and 105% of the expected benefit. Since the difference in the degree of expectation realization of *agent* is smaller, the relative deprivation sense between groups also decreases, and it can be seen from Figure 5 that the overall relative deprivation sense of the final *agent* group has significantly decreased and the difference in relative deprivation sense between *agents* has also narrowed, which shows the good robustness of the model. Combined with Table 3, it can be seen that the variation value of the total interest regulation among *agent* groups in the whole process of benefit redistribution is only 30, that is, the network freight platform only needs to deduct 30 from the commission revenue collected by the platform to greatly reduce the relative deprivation feeling among the network freight platform carrier groups, which indicates that the model has good interest regulation effect, that is, the model has good applicability in the problem of benefit redistribution among network freight platform carriers. This shows that the model has good interest regulation effect, that is, the model has good applicability in the problem of redistribution of interests of the carriers of the network freight platform.

According to Table 4 and the update of the indexes in Figure 6, it can be found that the mean value of the degree of expectation realization is stabilized at about 1.01 by the fifth iteration, and the variance keeps decreasing with the increase of the number of iterations, which indicates that the difference of the degree of expectation realization of benefits between groups is getting smaller. At the same time, according to Figure 7, it can be seen that the total benefit obtained by the *agent* group has a small variation, its maximum variation value is 73.33, and the total benefit of the *agent* group only increases by 30 after the redistribution process becomes stable compared with the initial one, which indicates that the cost of regulating the benefit redistribution behavior is low.

In addition, it can be seen from Figure 8 that the average value of relative deprivation of *agent* groups is decreasing, and there is a sharp peak slip during the first six iterations, which decreases from the initial 0.474 to 0.296, and then stabilizes at around this value. It can be seen that the difference in the degree of realization of expected benefits among the *agent* groups keeps shrinking in the process of iteration, and since the total benefits remain unchanged subsequently, it means that the benefits obtained by the *agents* after reallocation are closer to the expected benefits, and this result is also fully verified according to Figure 6. At the same time, the variance of the relative deprivation of the group is getting smaller and smaller in Figure 7, and finally tends to be stabilized, and the variance value is stabilized at about 0.0435 by Table 4, which means that the relative deprivation of the *agent* between groups is not much different, but since it has not yet fallen to the critical value of the relative deprivation of the group satisfaction of 0.1 and the number of iterations is less than *t_max_*, the benefit reallocation process still needs to be carried out. The benefit redistribution process still needs to be carried out. Finally, according to Table 4 and Figure 6, Figure 7 and Figure 8, it can be found that with the increase of the number of iterations, the changes of the subsequent indicators are very small or have reached a stable and constant state, which is because there is an upper limit of optimization in the problem scale, so the subsequent optimization cannot continue. For the problem studied in this paper, after the benefit redistribution process, the three measures of the variance value of the desired benefit realization degree, the group relative deprivation feeling and the group relative deprivation feeling variance of the platform carrier are significantly reduced, which indicates the good feasibility of the model in improving the fairness of benefit distribution.

### 5.3. Parameter Analysis

By adjusting some parameters, the applicability of the model to this problem is further verified. Assuming that the scale of profit distribution has changed now, the profit distribution in the final order distribution scheme shown in this paper is to be adopted as the initial profit distribution situation, and other parameters remain unchanged. The change trends of the operation results and key indicators in the distribution process are shown in Table 5, Table 6, Table 7 and Table 8 and Figure 9, Figure 10 and Figure 11, respectively.

From the results of Table 5, we can see that the degree of expected benefit realization and relative deprivation of the final *agent* group have increased and decreased respectively, and the change of benefits in the benefit redistribution process is only about 10, and the regulation cost is also low. In addition, combined with Figure 9, Figure 10 and Figure 11, it can be found that after the regulation of benefit redistribution, the expected benefit realization degree of *agent* is within the range of 1 ± 0.05, that is, the difference of expected benefit realization degree among the final *agent* group members is not too large, which is well reflected in the relative deprivation feeling, and it can be seen that the relative deprivation feeling changes from the blue area at the beginning of Figure 11 to the final brown area of the graph is significantly reduced, and it can be seen that the model has a good regulation effect on the distribution of benefits of different scales.

As can be seen from Table 6, as the number of iterations increases, the mean value of the desired benefit realization degree of the *agent* group is infinitely close to 1, and the mean value of the relative deprivation sense of the *agent* group is also decreasing, while the variance of the relative deprivation sense between groups is also decreasing, which, combined with the meanings of Equations (19), (20), (33) and (35), indicates that the whole coordination process of benefit distribution fairness is also getting higher. In other words, for the smaller scale benefit distribution problem, the indicators can still be dynamically regulated by the model algorithm and converge at the end, as initially envisioned. It can be seen that the model algorithm of this paper has good applicability for different scales of benefit distribution problems.

Finally, in order to further test the stability of this model, when following the above problem parameters, only the critical value σ of the relative deprivation sense of satisfaction of benefit distribution is changed, and the benefit distribution at σ = 0.3 and σ = 0.5, respectively, is examined. Since the initial distribution result at σ = 0.5 is already less than this critical value, there is no subsequent benefit reallocation for it, so no detailed description of its distribution result is made. σ = The benefit allocation results at 0.3 and the comparative analysis among the indicators when σ is taken as 0.1, 0.3 and 0.5, respectively, are shown in Table 7 and Table 8, respectively.

As can be seen from Table 7, the analysis of the two final allocation results comparing the model taking σ = 0.3 and σ = 0.1 shows that the difference between the mean value of the final expectation realization degree and the optimized value of the group relative deprivation sense when σ takes 0.1 and when σ takes 0.3 is not significant, but there is a large difference in the number of final completion allocation iterations between the two. The mean value of final relative deprivation at σ = 0.1 is smaller, which indicates that the final distribution of benefits at σ = 0.1 is more equitable than that at σ = 0.3, but the subsequent optimization at σ = 0.1 has reached the maximum and stabilized, so the mean value of group relative deprivation cannot move closer to 0.1 at the end. On the contrary, in the case of σ = 0.5, each optimization index is not ideal, because when the critical value is greater than the group relative deprivation sense mean in the initial state, the subsequent optimization also ceases to exist. Finally, when the three values of σ are compared with Table 8, it is found that when the value of σ is small, the final solution of benefit redistribution will be more fair, but the number of iterations will be more; on the contrary, when the value of σ is larger, the benefit redistribution will have an advantage in the number of iterations of redistribution, but the fairness of its distribution will be relatively reduced; in turn, the actual benefit distribution process can also confirm this point. In other words, different threshold criteria should be used for different groups to make the final benefit distribution scheme more effective, so as to minimize the degree of income inequality between groups and enhance the fairness of benefit distribution.

The practical implications of this paper are further analyzed by referring to the simulation experiments conducted by the researchers in [24,25]. In [24], the final adjustment result of the numerical simulation experiment makes the desired benefit realization degree of all agents stable around 1, while this paper also adjusts the desired benefit realization degree of all agents to around 1 and closer to 1. Since the total number of agents adjusted in this paper is more than that in [25], there is a higher lower limit of relative deprivation. However, the optimization results obtained from several numerical simulations still reduce the relative deprivation of the group by more than 30%, which has a better adjustment effect than the average reduction of about 10% in the 25 literatures. In combination with the study of Hailemariam et al. on the impact of income inequality on carbon emissions, it is clear that unfair income distribution increases per capita carbon emissions, which means that improving the fairness of benefit distribution can alleviate carbon pressure to a certain extent.

## 6. Conclusions

In the context of such a severe environmental situation nowadays, the whole country is facing high carbon pressure, and the trend of universal carbon trading is unstoppable. In addition, combined with the degree of impact of comprehensive carbon trading on different industries, this paper concludes that individual freight drivers are more severely affected by it. Therefore, in order to reduce the negative impact of income inequality on carbon emission reduction, this paper considers the fairness of benefit distribution from the perspective of relative deprivation, and thus improves the stability within the industry as much as possible. The study extends the group benefit distribution model based on the division of labor of ant colony, defines the network freight platform carrier drivers as individual ants and introduces contribution value, contribution rate and expected benefit realization degree to evaluate all individuals who complete the order transportation, so as to complete the setting of individual threshold and stimulation value. Then, we set the benefit realization ability according to the characteristics of individual ants, and calculate the relative deprivation feeling among groups, and coordinate the benefit distribution among carriers of the online freight platform in this way. The specific implementation of the solution problem in this paper is coded by MATLAB for the designed model and algorithm program, corresponding to the model and algorithm for changing the benefit distribution of the network freight platform. In this paper, the modeling and simulation of benefit allocation for the network freight platform carrier drivers are conducted, and it is found that the data obtained from the numerical simulation experiments for different size of benefit allocation scenarios show that the model has good flexibility, applicability and robustness. (1) The extended benefit distribution model constructed in this paper has a small range of benefit changes after the benefit redistribution process is completed, i.e., the cost of regulating the benefits among individuals is low for the online freight platform, and the overall relative deprivation of the entire group of online freight platform carriers can be reduced to a large extent with only a low cost of coordinating benefits for the platform; (2) After the regulation, the desired benefits of the network freight platform carrier drivers are infinitely close to 1, and the relative deprivation is significantly reduced, which indicates that the fairness of the benefit distribution is improved, and shows the good applicability and robustness of the model; (3) After changing the parameters several times, we finally found that the change trend of various indicators and the final simulation results are more consistent with the real benefit distribution problem, and the overall coordination effect of the simulation results is not very different, which shows the reasonableness of the model construction and the stability of the model from the side.

In summary, the research in this paper can better reduce the negative impact of income inequality on the stability of network freight platform and carbon emission, but there are still shortcomings in the research process. In the study of the distribution of benefits among carriers of network freight platforms, some real data are missing, and some simulated data satisfying random distribution are used to conduct numerical experiments to ensure the non-chance of experiments, so some calculation results have some limitations. In this paper, we consider the distribution of interests among the carrier groups in the network freight platform, and do not explore the distribution of interests between the network platform and the actual carriers, and we can cooperate with the network freight enterprises to obtain more detailed data and conduct a more comprehensive study on the distribution of interests in the network freight platform. The model and solution algorithm designed in this paper have been verified by several numerical experiments, and the results show that they provide optimization support for the fair benefit distribution between the carriers of the network freight platform, which can provide some reference for the formulation of the relevant benefit distribution system.

## Figures and Tables

**Figure 1 ijerph-19-15031-f001:**
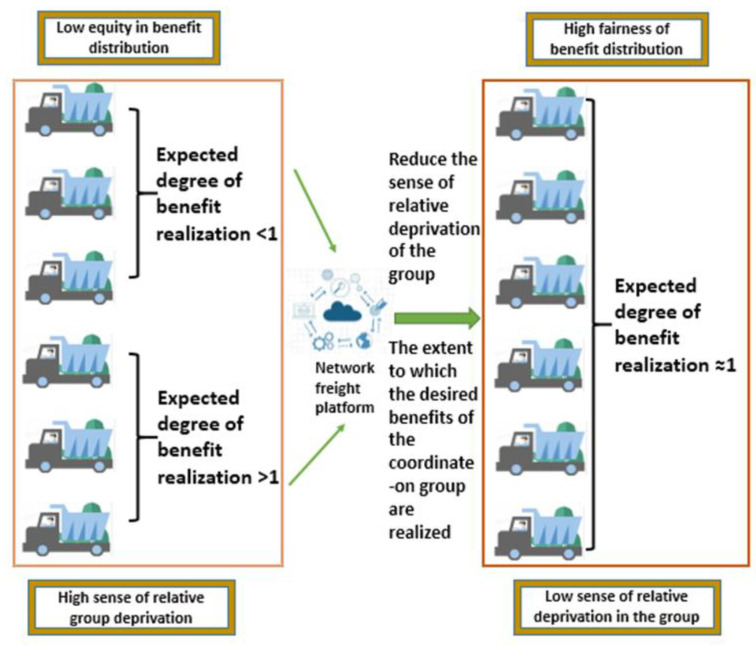
Network freight carrier driver benefit redistribution scenario.

**Figure 2 ijerph-19-15031-f002:**
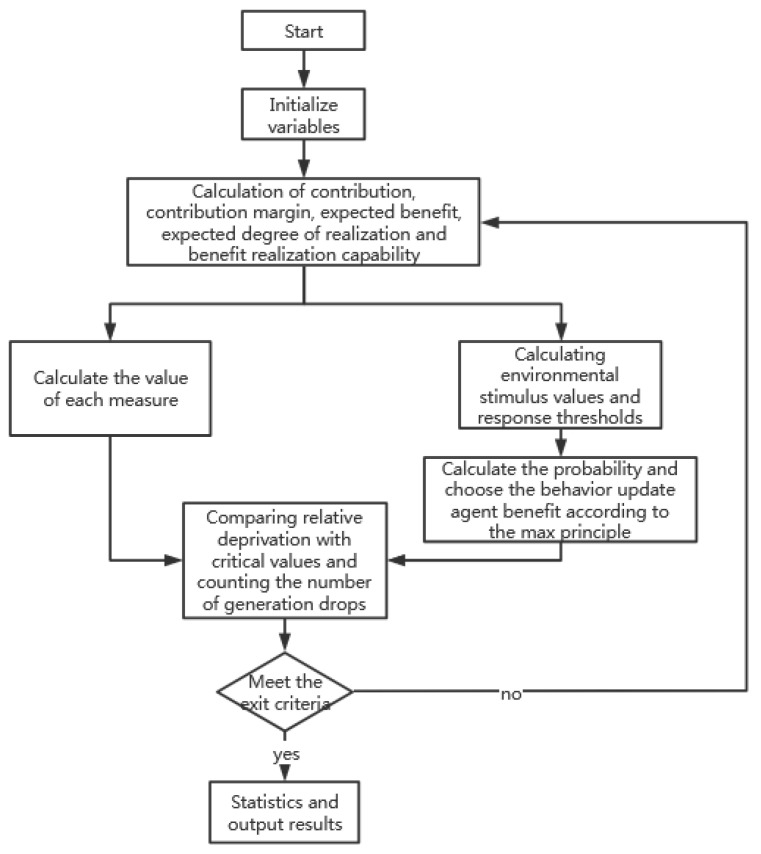
Flow chart of the extended benefit allocation algorithm.

**Figure 3 ijerph-19-15031-f003:**
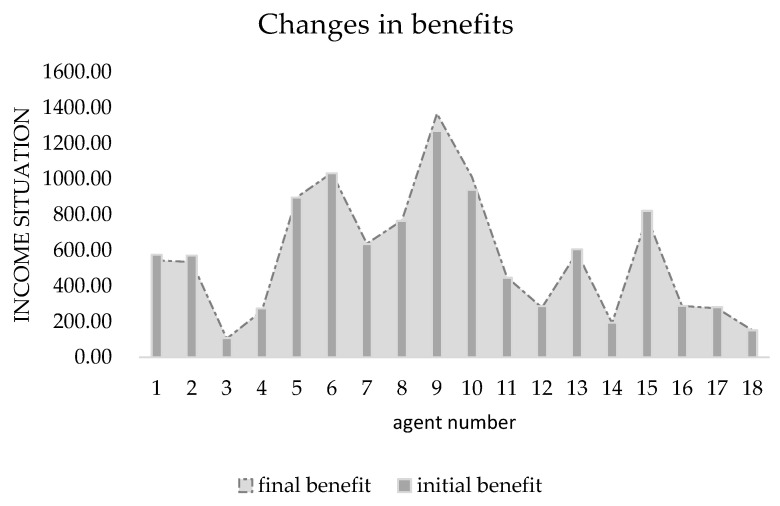
Changes of *agent* benefits.

**Figure 4 ijerph-19-15031-f004:**
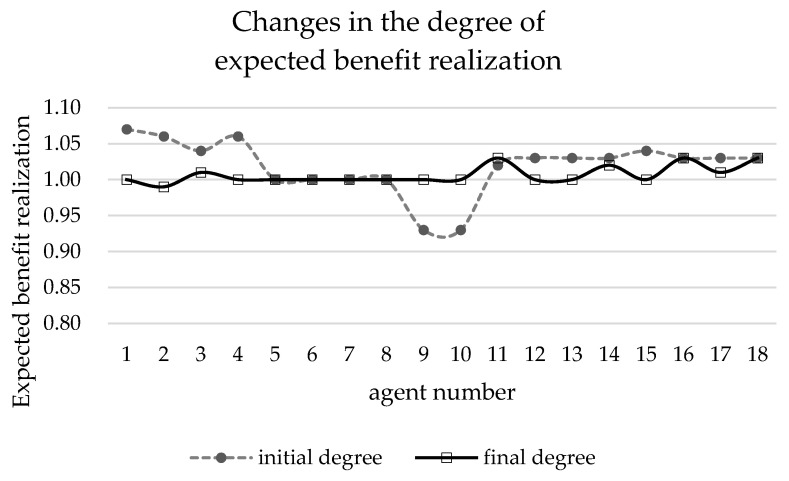
Changes in the degree of *agent*.

**Figure 5 ijerph-19-15031-f005:**
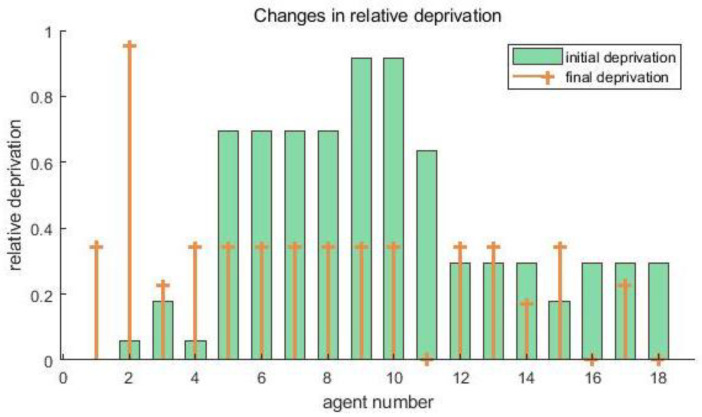
Changes in relative deprivation in *agent*.

**Figure 6 ijerph-19-15031-f006:**
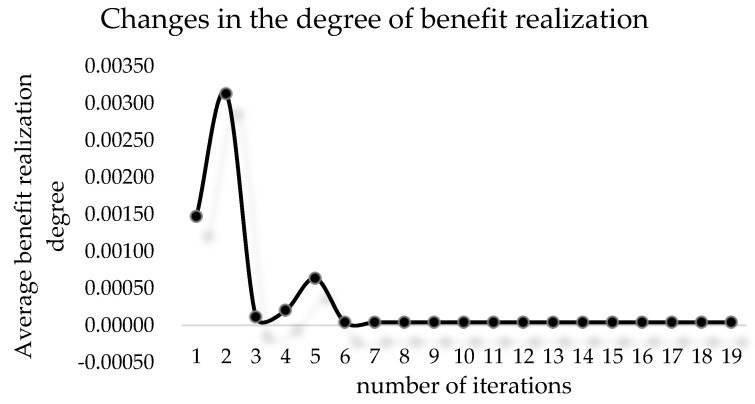
Update on the variance of the degree of interest.

**Figure 7 ijerph-19-15031-f007:**
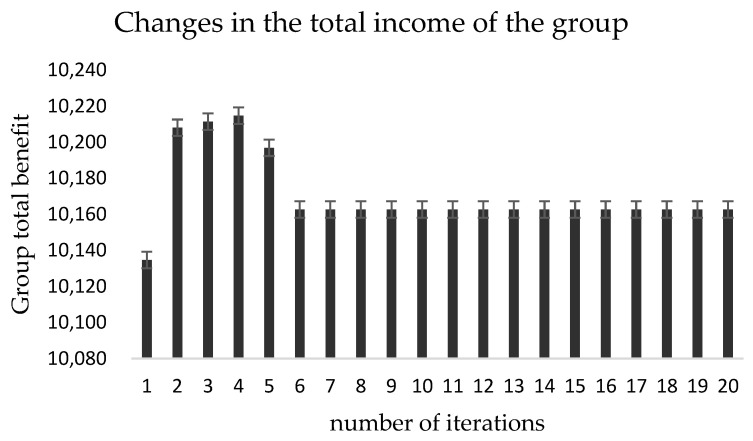
Update of total group benefits.

**Figure 8 ijerph-19-15031-f008:**
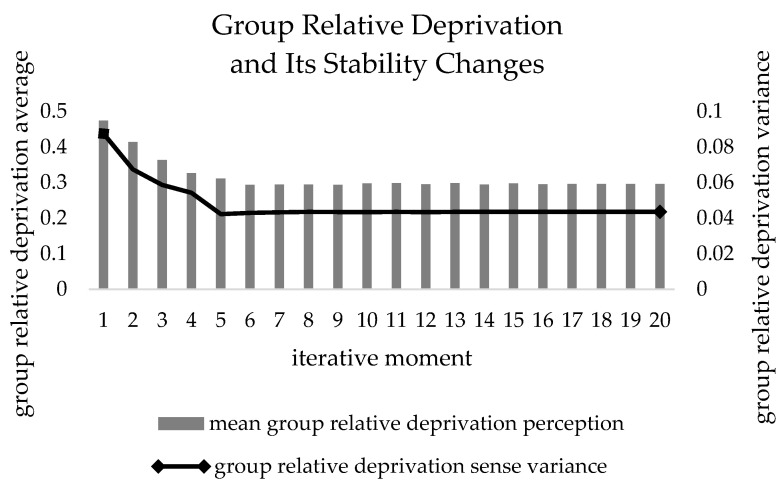
Relative deprivation and its variance update.

**Figure 9 ijerph-19-15031-f009:**
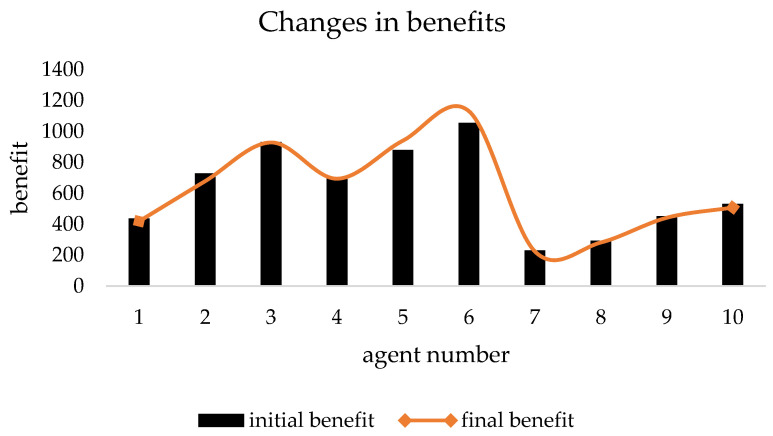
Changes in *agent* benefits.

**Figure 10 ijerph-19-15031-f010:**
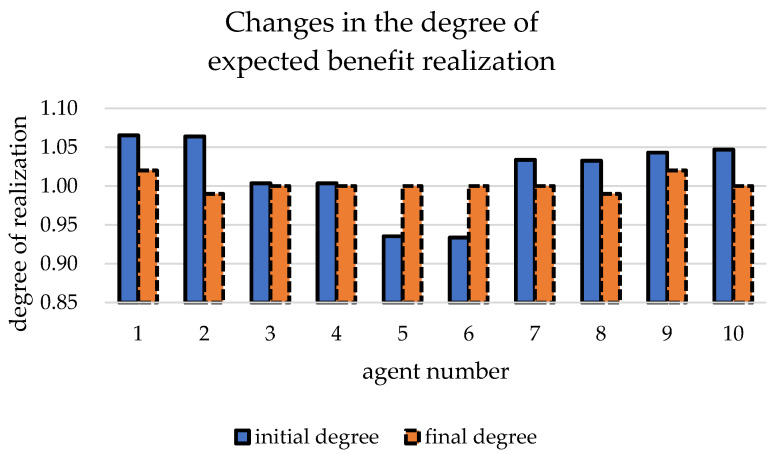
Changes in *agent* realization.

**Figure 11 ijerph-19-15031-f011:**
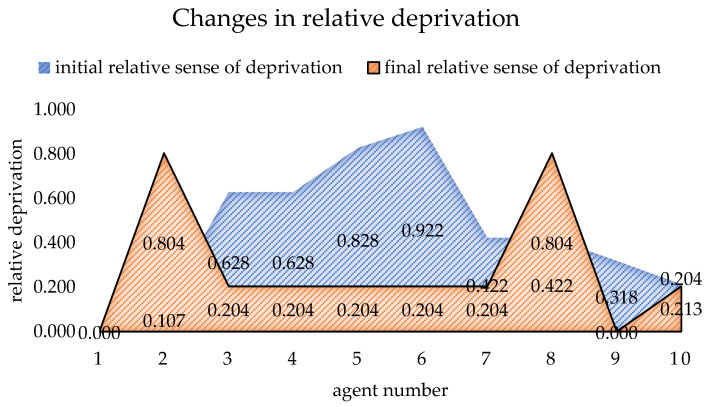
Change in agent's relative deprivation after adjusting parameters.

**Table 1 ijerph-19-15031-t001:** Variable description.

Variable Name	Meaning
*AC_i_*	*agenti* contribution
*ac_i_*	*agenti* contribution rate to groups
*AG_i_*(*t*)	*agenti* benefits divided in the process of benefit distribution at time t
*Q_i_*	*agenti* expectation of benefits in the process of benefit distribution
*A_i_*(*t*)	*agenti* the degree of expected benefit is realized at time t
*RD_i_*(*t*)	*agenti* relative deprivation at time t
*AE_i_*(*t*)	*agenti* ability to realize benefits at time t
$S_{i}{}^{+}\left(t\right)$	*agenti* increased environmental stimulus value of interest induced at time t
$S_{i}{}^{-}\left(t\right)$	*agenti* reduced environmental stimulus value of interest induced at time t
$\theta_{i}{}^{+}\left(t\right)$	*agenti* response threshold for the increased benefit of the response at time t
$\theta_{i}{}^{-}\left(t\right)$	*agenti* response threshold for reducing the benefit of the response at time t
$\varepsilon_{i}\left(t\right)$	*agenti* change of interest at time t
*σ*	The threshold value of relative deprivation when the benefit distribution is satisfied
*ν_i_*	*agenti* maximum carrying capacity
*c_i_*	*agenti* unit cost of transportation
*μ_i_*	*agenti* amount per unit of carbon emissions of
*w*	Carbon tax
*α*	Transportation cost conversion factor
*β*	Carbon emission environmental cost conversion coefficient
$\xi_{1}$	Interest regulation increment
$\xi_{2}$	Interest adjustment is negative increment
*ρ*	Benefit realization ability conversion coefficient
*k*	Transformation coefficient of the threshold value

**Table 2 ijerph-19-15031-t002:** Initial *agent* revenue and vehicle model situation.

New *agent* Code Name	*agent* Motorcycle Type	Proceeds from the Initial Distribution	Benefit Realization Ability
*agent*1	Small van	575.24	3.33
*agent*2	Small van	570.24	3.33
*agent*3	giant flat truck	107.52	3.33
*agent*4	Small van	273.76	3.33
*agent*5	Medium van	895.44	3.00
*agent*6	Medium van	1031.90	3.00
*agent*7	Large van	634.92	3.00
*agent*8	Medium plate car	764.52	3.00
*agent*9	Medium van	1269.48	2.67
*agent*10	giant flat truck	939.60	2.67
*agent*11	Medium plate car	446.00	3.00
*agent*12	Large van	286.74	3.00
*agent*13	Medium van	605.36	3.00
*agent*14	giant flat truck	193.12	3.00
*agent*15	Medium plate car	820.80	3.06
*agent*16	Small van	288.00	3.06
*agent*17	giant flat truck	280.72	3.06
*agent*18	Medium plate car	151.28	3.06

**Table 3 ijerph-19-15031-t003:** Results of benefit distribution.

*agent*	Contribution Value AC	Initial Income AG	Finally Gains AG’	Initial Expectations A	Final Expectation A’	Initial Deprivation RD	Final Sense of Deprivation RD’
1	393.226	575.24	543.35	1.07	1.00	0.000	0.341
2	390.679	570.24	534.44	1.06	0.99	0.059	0.952
3	74.924	107.52	104.56	1.04	1.01	0.177	0.228
4	187.489	273.76	259.07	1.06	1.00	0.059	0.341
5	649.388	895.44	897.31	1.00	1.00	0.693	0.341
6	748.265	1031.90	1033.94	1.00	1.00	0.693	0.341
7	461.478	634.92	637.66	1.00	1.00	0.693	0.341
8	554.641	764.52	766.39	1.00	1.00	0.693	0.341
9	989.000	1269.48	1366.58	0.93	1.00	0.915	0.341
10	732.100	939.60	1011.60	0.93	1.00	0.915	0.341
11	315.394	446.00	448.88	1.02	1.03	0.636	0.000
12	201.507	286.74	278.44	1.03	1.00	0.293	0.341
13	425.209	605.36	587.55	1.03	1.00	0.293	0.341
14	136.333	193.12	192.15	1.03	1.02	0.293	0.172
15	569.373	820.80	786.75	1.04	1.00	0.177	0.341
16	202.391	288.00	288.05	1.03	1.03	0.293	0.000
17	196.617	280.72	274.40	1.03	1.01	0.293	0.228
18	106.455	151.28	151.51	1.03	1.03	0.293	0.000
total	7334.469	10,134.64	10,162.64	18.30	18.244	8.540	5.326

**Table 4 ijerph-19-15031-t004:** Updates of relevant indicators.

Iterations t	Value of Variance of Expected Benefit Realization	Group Total Income	Group Deprivation Average Value	Group Deprivation Variance
1	0.00147	10,134.64	0.474	0.0874
2	0.00313	10,207.97	0.414	0.0674
3	0.00011	10,211.31	0.363	0.0586
4	0.00020	10,214.64	0.326	0.0542
5	0.00064	10,196.85	0.311	0.0422
6	0.00004	10,162.64	0.293	0.0429
7	0.00004	10,162.64	0.294	0.0432
8	0.00004	10,162.64	0.294	0.0435
9	0.00004	10,162.64	0.293	0.0434
10	0.00004	10,162.64	0.297	0.0433
11	0.00004	10,162.64	0.298	0.0435
12	0.00004	10,162.64	0.295	0.0433
13	0.00004	10,162.64	0.298	0.0435
14	0.00004	10,162.64	0.294	0.0435
15	0.00004	10,162.64	0.297	0.0435
16	0.00004	10,162.64	0.295	0.0435
17	0.00004	10,162.64	0.296	0.0435
18	0.00004	10,162.64	0.296	0.0435
19	0.00004	10,162.64	0.296	0.0435
20	0.00004	10,162.64	0.296	0.0435

**Table 5 ijerph-19-15031-t005:** Result of benefit distribution after adjusting parameters.

*Agent*	Contribution Value AC	Initial Income AG	Ultimate Yield AG’	Initial Expectations A	Final Expectation A’	Initial Deprivation RD	Final Sense of Deprivation RD’
1	299.51	438.00	419.34	1.07	1.02	0.000	0.000
2	499.24	729.00	678.43	1.06	0.99	0.107	0.804
3	676.69	932.20	928.87	1.00	1.00	0.628	0.204
4	505.95	697.00	694.49	1.00	1.00	0.628	0.204
5	685.45	880.00	940.89	0.94	1.00	0.828	0.204
6	823.94	1056.00	1130.99	0.93	1.00	0.922	0.204
7	163.36	231.80	224.23	1.03	1.00	0.422	0.204
8	207.71	294.40	282.26	1.03	0.99	0.422	0.804
9	315.95	452.40	442.36	1.04	1.02	0.318	0.000
10	370.18	532.00	508.13	1.05	1.00	0.213	0.204
total	4547.98	6242.80	6249.99	10.16	10.02	4.488	2.832

**Table 6 ijerph-19-15031-t006:** Update of relevant indicators after adjustment of parameters.

Iterations t	Average Value of the Expected Benefit Realization Degree	Group Total Income	Group Deprivation Average Value	Group Deprivation Variance
1	1.00	6242.80	0.448	0.0008
2	1.02	6282.80	0.392	0.0018
3	1.01	6302.80	0.365	0.0006
4	1.00	6322.80	0.339	0.0013
5	1.02	6249.99	0.342	0.0017
6	1.01	6249.99	0.340	0.0006
7	1.00	6249.99	0.283	0.0001
8	1.02	6249.99	0.283	0.0001
9	1.01	6249.99	0.283	0.0001
10	1.00	6249.99	0.283	0.0001
11	1.02	6249.99	0.283	0.0001
12	1.01	6249.99	0.283	0.0001
13	1.00	6249.99	0.283	0.0001
14	1.02	6249.99	0.283	0.0001
15	1.01	6249.99	0.283	0.0001
16	1.00	6249.99	0.283	0.0001
17	1.00	6249.99	0.283	0.0001
18	1.00	6249.99	0.283	0.0001
19	1.00	6249.99	0.283	0.0001
20	1.00	6249.99	0.283	0.0001

**Table 7 ijerph-19-15031-t007:** Profit distribution results for σ = 0.2.

*Agent*	Contribution Value AC	Initial Income AG	Ultimate Yield AG’	Initial Expectations A	Final Expectation A’	Initial Deprivation RD	Final Sense of Deprivation RD’	Number of Iterations on Completion
1	299.51	438.00	417.53	1.07	1.02	0.000	0	/
2	499.24	729.00	680.86	1.06	0.99	0.107	0.808	/
3	676.69	932.20	941.52	1.00	1.01	0.628	0.102	/
4	505.95	697.00	703.97	1.00	1.01	0.628	0.102	/
5	685.45	880.00	945.53	0.94	1.01	0.828	0.102	/
6	823.94	1056.00	1146.84	0.93	1.01	0.922	0.102	/
7	163.36	231.80	227.30	1.03	1.01	0.422	0.102	/
8	207.71	294.40	288.68	1.03	1.01	0.422	0.102	/
9	315.95	452.40	435.00	1.04	1	0.318	0.708	/
10	370.18	532.00	501.60	1.05	0.99	0.213	0.808	/
total	4547.98	6242.80	6288.84	10.16	10.06	4.488	2.936	5

**Table 8 ijerph-19-15031-t008:** Comparison of relevant indicators for different cut-off values.

	Measuring Indicators	Change in Total Interests	Final Expected Degree of Realization Means	Final Mean Sense of Relative Deprivation	Final Relative Deprivation Sense of Variance	Final Number of Assignment Iterations Is Completed
The Critical Value σ						
0.1		7.19	1.00	0.2832	0.1656	20
0.3		59.20	1.01	0.2936	0.1552	5
0.5		0	1.01	0.4488	0	1

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
