# Peer review of "Profit Allocation Problem and Algorithm of Network Freight Platform under Carbon Trading Background"

_ijerph, 2022, doi:10.3390/ijerph192215031_

Round 1

Reviewer 1 Report

The authors state that the current research promotes the development of low-carbon freight transport takes low-carbon emissions as an environmental constraint, and improves the fixed threshold response model based on the division of labor of the ant colony. 

The research background and research problem of this paper are interesting. The presentation quality is reasonable, and the narrative is clear and understandably to me. Nevertheless, some important points have to be clarified and improved:

1) In the introduction part, please be more specific about the paper's contributions to knowledge and theory.

2) In the introduction part, it is better to provide a figure to demonstrate the problem studied.

3) Please comment on how can the values of weight coefficient of each index j in equation 1 be chosen, as well as the weight coefficient of the range of regulating threshold in equations 8 and 9.

4) The content for using MATLAB R2018A programming is not enough. Please provide more details about the algorithm implementation. It would be better to describe the algorithm using flowcharts.

5) How the authors tuned the model parameters in the problem description and parameter setting part?

6) In your discussion section, please link your numerical study results with a broader and deeper literature review.

7) The research gaps identified by the authors are not well addressed. Please be more specific on the scientific contribution of this paper.

8) Please cite and give more comments about the table in the appendix.

9) Please revise and correct the sentences below:

L26 : allocation, We will optimize

L202-203, 207, 247, 271, 283, 315, 316, 332 and more : please check spaces errors in text

L205: do you mean agent i?

L305-306: so the research object in this chapter is to obtain the benefit distribution agent

L467: the simulation experiment in this chapter is carried

L619: (1) This chapter is to build

L613: behavior choice, Respectively

Author Response

The authors state that the current research promotes the development of low-carbon freight transport takes low-carbon emissions as an environmental constraint, and improves the fixed threshold response model based on the division of labor of the ant colony. 
The research background and research problem of this paper are interesting. The presentation quality is reasonable, and the narrative is clear and understandably to me. Nevertheless, some important points have to be clarified and improved:
1)    In the introduction part, please be more specific about the paper's contributions to knowledge and theory.
Response: added a redundancy of the contribution made by the article
2) In the introduction part, it is better to provide a figure to demonstrate the problem studied.
Response: the problem scenario diagram has been added and explained
3) Please comment on how can the values of weight coefficient of each index j in equation 1 be chosen, as well as the weight coefficient of the range of regulating threshold in equations 8 and 9.
Response: equation 1 is the contribution value, the purpose of which is to quantify the contribution made by an individual in the task environment. Since the object of this paper is the network freight platform carrier driver, and this paper is the regulation of the benefit distribution after the order is completed under the background constraint containing carbon tax, the contribution actually made by the network freight carrier driver is the transportation cost and carbon cost borne by him. Since there is a difference between the unit transportation cost and the unit carbon cost, in order to reduce the difference in magnitude and to highlight the importance of the carbon constraint, this paper uses the weight coefficient to raise the magnitude of the two to the same level. The weight coefficients in Equation 8 and Equation 9 are set to adjust the threshold magnitude to the same level as the stimulus value, and the approximate numerical magnitude of the stimulus value can be obtained by bringing the value into the calculation, thus setting up a range of weight coefficients for the threshold value, making the calculation of the probability of individual choice behavior into a reasonable interval range.
4) The content for using MATLAB R2018A programming is not enough. Please provide more details about the algorithm implementation. It would be better to describe the algorithm using flowcharts.
Response: algorithm flow chart has been added
5) How the authors tuned the model parameters in the problem description and parameter setting part?
Response: in the problem description, the indicators related to benefit distribution are extracted, and the indicators related to order transportation in the network freight platform are used as the base variables of the model; the parameter settings in the model are based on the comparison of the regulation effect when different parameters are set in the simulation experiment, and the parameter settings with better effect are selected, and further parameter settings are developed for problems of different scales and dimensions to verify the effectiveness and applicability of the model.
6) In your discussion section, please link your numerical study results with a broader and deeper literature review.
Response: added some links between the research results and the previous review in the Discussion and Conclusion section
7) The research gaps identified by the authors are not well addressed. Please be more specific on the scientific contribution of this paper.
Response: additional explanation has been provided at the conclusion
8) Please cite and give more comments about the table in the appendix.
Response: additional descriptions have been added
9) Please revise and correct the sentences below:
L26 : allocation, We will optimize
Response: modifications have been made
L202-203, 207, 247, 271, 283, 315, 316, 332 and more : please check spaces errors in text
Response: the above space error has been corrected and the full statement problem has been verified
L205: do you mean agent i?
Response: yes, the translation is wrong, the i is missing
L305-306: so the research object in this chapter is to obtain the benefit distribution agent
L467: the simulation experiment in this chapter is carried
L619: (1) This chapter is to build
Response: all of the above changed chapter to paper
L613: behavior choice, Respectively
Response: modifications have been made

Reviewer 2 Report

This paper presents a dynamic benefit distribution optimization model oriented to the sense of relative deprivation in a freight network platform under carbon trading. It is based on indicators such as contribution value, degree of expectation realization and relative sense of deprivation.

The topic is interesting, but the manuscript needs to be improved.

The issue of carbon cost in freight transportation, described in the Introduction section, should be defined in its entirety, as the most recent scientific studies highlight other aspects. The externalities caused by the fright transport is not only related to emissions, and there are different methodologies to evaluate them in economic form, that should be mentioned:

         An Optimization-Based Approach to Evaluate the Operational and Environmental Impacts of Pick-Up Points on E-Commerce Urban Last-Mile Distribution: A Case Study in São Paulo, Brazil

         Optimization of Goods Relocation in Urban Store Networks with an Incentive Strategy

         Crowdshipping in last mile deliveries: Operational challenges and research opportunities

There are some unclear parts:

-       Lines 40-41. The sentence needs to be rephrased better because it is not understandable;

-       Several concepts are repeated more times in the text without bringing more information but only the same concept. It is appropriate to delate where not necessary (i.e. lines 55-60 and 76; line 156);

-       Lines 69-70. The sentence need to be rephrased, there is a mistake;

-       Lines 69-75, 221-225. The sentences are too long. It is necessary to divide them;

-       Lines 276-278. The sentence need to be rephrased: what does 1=0 mean?

-       Line 284. What does “interests of interests …. redistribution interests” mean?

-       Line 554. What does “Figure 7 – 9 to 4 - 11” mean? The total number of figures in the manuscript is 9.

-       Lines 554-558. The sentence need to be rephrased. It is unclear.

-       Line 559. What does “Figure 4 - 11” mean? The total number of figures in the manuscript is 9.

-       Lines 608-619. The sentence need to be rephrased. It is unclear.

-       Line 619. What does “chapter” mean?

One very important aspect of the manuscript is related to the form. A thorough reading is necessary, as the number of mistakes and typos is very high. In this form, the work cannot be published.

Only as an example, there are:

-       Sentences with repetitive words that make reading uncomfortable (lines 

-       sentences in which punctuation is incorrect, with semicolons or comma instead of period (lines 26, 226, 293, 398, 431, 456, 523, 578, 613)

-       sentences with incorrect spacing (lines 257, 259, 271, 315, 316, 332, 339, 340, 346, 355, 356, 373, 391, 425, 426)

-       incorrect capitalized words (line 445, 458, 475

Author Response

This paper presents a dynamic benefit distribution optimization model oriented to the sense of relative deprivation in a freight network platform under carbon trading. It is based on indicators such as contribution value, degree of expectation realization and relative sense of deprivation.
The topic is interesting, but the manuscript needs to be improved.
The issue of carbon cost in freight transportation, described in the Introduction section, should be defined in its entirety, as the most recent scientific studies highlight other aspects. The externalities caused by the fright transport is not only related to emissions, and there are different methodologies to evaluate them in economic form, that should be mentioned:
         An Optimization-Based Approach to Evaluate the Operational and Environmental Impacts of Pick-Up Points on E-Commerce Urban Last-Mile Distribution: A Case Study in São Paulo, Brazil
         Optimization of Goods Relocation in Urban Store Networks with an Incentive Strategy
         Crowdshipping in last mile deliveries: Operational challenges and research opportunities
Response: the relevant parts of the above mentioned references to the literature have been added to the introduction, as follows:and the externalities caused by goods transportation are not only related to carbon emis-sions, Pourrahmani and Jaller [3] in their study for crowdshipping in last mile deliveries found that there is an impact of evaluating services on service participants, which pointed out that appropriate compensation schemes can lead to high capital efficiency and sus-tainable operations. And Silvestri et al [4] in their study for the optimization of networked goods migration in urban stores pointed out that a shared information based model of networked goods migration can reduce the pollution generated during transportation and thus its associated externalities. Masteguim and Cunha [5], on the other hand, conducted a study on the impact of pickup points on last-mile delivery and showed that reasonable delivery points can circumvent more waste of resources.

There are some unclear parts:
-       Lines 40-41. The sentence needs to be rephrased better because it is not understandable;
Response: modified and adjusted
-       Several concepts are repeated more times in the text without bringing more information but only the same concept. It is appropriate to delate where not necessary (i.e. lines 55-60 and 76; line 156);
Response: lines 55-76 are translation errors, duplicate statements, and have been deleted
-       Lines 69-70. The sentence need to be rephrased, there is a mistake;
Response: already deleted
-       Lines 69-75, 221-225. The sentences are too long. It is necessary to divide them;
Response: lines 69-75 have been deleted and lines 221-225 have been adjusted
-       Lines 276-278. The sentence need to be rephrased: what does 1=0 mean?
Response: translation error, removed
-       Line 284. What does “interests of interests …. redistribution interests” mean?
Response: written error, actually means:considering that the interests of the interest subject will change in the process of ben-efit redistribution, the demand of the interest subject desiring benefit redistribution will gradually decrease at this time
-       Line 554. What does “Figure 7 – 9 to 4 - 11” mean? The total number of figures in the manuscript is 9.
Response: the translation was incorrect and has been changed to Figure 9
-       Lines 554-558. The sentence need to be rephrased. It is unclear.
-       Line 559. What does “Figure 4 - 11” mean? The total number of figures in the manuscript is 9.
Response: translation error, has been changed, the actual is Figure 9
-       Lines 608-619. The sentence need to be rephrased. It is unclear.
Response: modifications have been made
-       Line 619. What does “chapter” mean?
Response: the original meaning of this article has been corrected due to a translation error
One very important aspect of the manuscript is related to the form. A thorough reading is necessary, as the number of mistakes and typos is very high. In this form, the work cannot be published.
Only as an example, there are:
-       Sentences with repetitive words that make reading uncomfortable (lines 
-       sentences in which punctuation is incorrect, with semicolons or comma instead of period (lines 26, 226, 293, 398, 431, 456, 523, 578, 613)
-       sentences with incorrect spacing (lines 257, 259, 271, 315, 316, 332, 339, 340, 346, 355, 356, 373, 391, 425, 426)
-       incorrect capitalized words (line 445, 458, 475
Response: the above errors have been fixed and the full text has been checked for translation errors

Round 2

Reviewer 1 Report

As for the updated revision conducted by the authors, the improvement can be observed in different sections. The authors have responded to the major concerns that I raised in the previous review round. The revised manuscript sounds more legible than before. Here are some minor comments:

1) I invite the authors to be clearer about their response for my previous concern n°6) In your discussion section, please link your numerical study results with a broader and deeper literature review.
Authors response: added some links between the research results and the previous review in the Discussion and Conclusion section.

2) A thorough reading is necessary, as the number of sentences with incorrect spacing is still very high. 

Author Response

As for the updated revision conducted by the authors, the improvement can be observed in different sections. The authors have responded to the major concerns that I raised in the previous review round. The revised manuscript sounds more legible than before. Here are some minor comments:

1) I invite the authors to be clearer about their response for my previous concern n°6) In your discussion section, please link your numerical study results with a broader and deeper literature review.
Authors response: added some links between the research results and the previous review in the Discussion and Conclusion section. And on the basis of the original, the results of the numerical study are compared and analyzed with the results of previous studies, and the significance of the study is explained in the context of this paper

2) A thorough reading is necessary, as the number of sentences with incorrect spacing is still very high.
Authors response: the inter-sentence spacing issue was rechecked and the inter-sentence spacing has been readjusted.

Reviewer 2 Report

The paper has been greatly improved in this new version.

Author Response

The paper has been greatly improved in this new version.

Author response:Thanks